# Automatic RTL Generation Tool of FPGAs for DNNs

Seojin Jang [1], Wei Liu [2], Sangun Park [3] and Yongbeom Cho [1,2,*]

1 Department of Electrical and Electronics Engineering, Konkuk University, Seoul 05029, Korea; seojinygud@konkuk.ac.kr
2 Deep ET, Seoul 05029, Korea; liuwei1108@deep-et.com
3 Samsung Electronics, Suwon 16677, Korea; kos8108@naver.com
* Correspondence: ybcho@konkuk.ac.kr

**Abstract:** With the increasing use of multi-purpose artificial intelligence of things (AIOT) devices, embedded field-programmable gate arrays (FPGA) represent excellent platforms for deep neural network (DNN) acceleration on edge devices. FPGAs possess the advantages of low latency and high energy efficiency, but the scarcity of FPGA development resources challenges the deployment of DNN-based edge devices. Register-transfer level programming, hardware verification, and precise resource allocation are needed to build a high-performance FPGA accelerator for DNNs. These tasks present a challenge and are time consuming for even experienced hardware developers. Therefore, we propose an automated, collaborative design process employing an automatic design space exploration tool; an automatic DNN engine enables the tool to reshape and parse a DNN model from software to hardware. We also introduce a long short-term memory (LSTM)-based model to predict performance and generate a DNN model that suits the developer requirements automatically. We demonstrate our design scheme with three FPGAs: a zcu104, a zcu102, and a Cyclone V SoC (system on chip). The results show that our hardware-based edge accelerator exhibits superior throughput compared with the most advanced edge graphics processing unit.

**Keywords:** convolutional neural network (CNN); deep learning; hardware/software co-design; FPGA

## 1. Introduction

Recent studies have shown that field-programmable gate arrays (FPGA) are promising candidates for deep neural network (DNN) implementation [1–5]. A DNN can be integrated via hardware, rather than via an existing central processing unit (CPU) or graphics processing unit (GPU), thus improving latency and reducing energy consumption. These characteristics exemplify FPGAs for DNN-based applications in cloud and edge computing; as a result, FPGAs have been rapidly adopted for DNN acceleration. Internet of things (IOT) applications have stringent requirements in the fields of automatic driving, safety, and monitoring; complex DNN models must produce quality results with minimal delay and power consumption, while not exceeding resource constraints [6].

Embedded FPGAs are the most attractive candidate to integrate machine learning with IOT [7] because they are energy efficient and affordable. However, development resource scarcity makes the design and deployment of FPGA DNN accelerators challenging.

Our study alleviates several challenges in the field with emphasis on the following main ideas:

(1) This study proposes DNN implementation via an auto-generated automation tool, which maps the DNN design process from a deep learning framework to FPGA. As the structure of DNNs changes rapidly, it is difficult for their hardware design to keep up with the software design. Therefore, our proposition allows users to perform resource allocation and optimization during register-transfer level (RTL) design when deploying a DNN on FPGAs.



(2)   A DNN model was generated according to developer performance guidelines during the design process. An automatic DNN model optimization engine is proposed based on this process, by implementing a long short-term memory (LSTM) that can effectively generate DNN models that meet the performance requirements of FPGA design.

(3)   Highly optimized RTL network components can be automatically generated for building DNN layers. Since different FPGA manufacturers use different intellectual property (IP) cores for multiplication in digital signal processing (DSP), we propose a multiplier optimization for DNN processing elements (PEs) with improved energy consumption. The engine can be configured to provide the best performance within the constraints of the FPGA.

## 2. Related Work

Designing a DNN model and FPGA accelerator requires meticulous research, which is often carried out independently. Machine learning experts either manually design DNNs or employ automatic design via a recurrent neural network (RNN) and reinforcement learning. In FPGA acceleration, traditional and Winograd convolutions are combined to run DNNs on embedded FPGAs [8]. Aydonat et al. showed that the Winograd-based solution could be implemented on FPGAs, reducing the need for multiplication in a DNN [9].

Studies involving platform-based DNN search methods [10,11] improve upon previously published results. However, the authors of these studies only considered DNN inference delay on the CPU and GPU, neglecting DNN inference delay on FPGAs.

New techniques such as quantization [8,12] and model compression [13,14] are also used in the FPGA DNN accelerator to reduce the size of the DNN model and diminish latency in the DNN inference process. However, these methods may be limited by the DNN model itself, as not all DNN models can be compressed. In addition, the compressed DNN model does not necessarily meet the real-time performance constraints of target artificial IOT (AIOT) platforms.

Automation tools to quickly deploy DNNs into FPGAs have been investigated by other researchers [15]. They adopted a unified RTL design for the convolution (conv) layer to permit different network configurations, but the fully connected (FC) layer was not implemented on the FPGA. The proposed RTL template mapped DNN onto FPGAs automatically. However, the DLL used a unified computing unit (for example, a fixed-size computing engine), which yields reduced computing performance when compared with a dedicated design for different networks [16,17]. In these studies, the high-level synthesis (HLS) template was used to map DNNs to FPGAs automatically, significantly improving the RTL size [18].

## 3. Materials and Methods

### 3.1. Proposed Automation Flow

We propose a space exploration model to find the optimal accelerator design that balances the DNN network's constraints and the FPGA's performance. As shown in Figure 1, we generated the DNN accelerator via a three-step procedure that included a network analyzer, required-performance-based PE optimization, and hardware description language (HDL) auto-generation. The steps are described as follows:

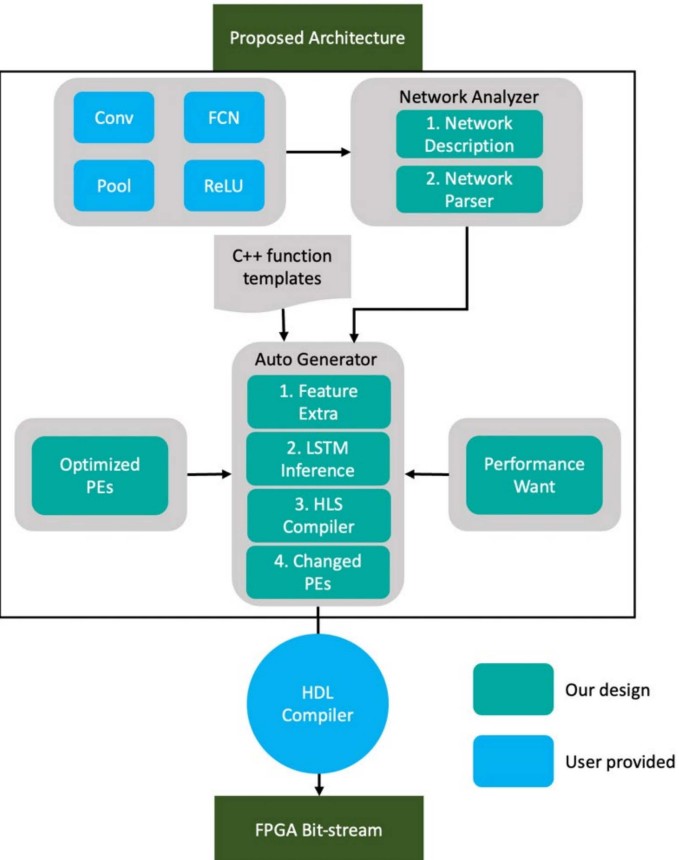

**Figure 1.** Proposed system.

(1) Developers first use a deep learning open neural network exchange (ONNX) framework to design and train the DNN target network during the design process. After training, the DNN network definition file is passed to the DNN network analyzer [19].

(2) A network analysis decomposes network layers from the network model, such as the conv, pooling, and FC layers. Then, it maps them to the HLS template. The network analyzer retains the operation logic and bit size in the conv, pooling, and FC layers to ensure that logic remains as designed.

(3) In the third step, the LSTM-based generator automatically designs the optimized DNN HLS.

We optimized the DNN design in two ways. First, we redesigned the PE so that energy consumption is optimized in the automatically generated neural network. The proposed PE also supports N-bit calculations, permitting the automatic quantization of the network into N-bits. Second, we implemented feature value extraction in the HLS C/C++ inference code and used this feature value to train the LSTM. This automatically generates the DNN-RTL design with the highest degree of parallelism.

### 3.2. N-Bit Vedic Multiplier for DNN PEs

PEs are an essential tool in DNN operations. However, there are two limitations when using an embedded multiplier or embedded DSP block to construct a PE: PE size is fixed, and PE availability is limited. In addition, they have a fixed position on the FPGA itself and often have bit size asymmetry (the $18 \times 25$ multiplier of Xilinx FPGAs, for example) [20]. Therefore, we designed a Vedic multiplier-based N-bit processing element to overcome the limitations above. Vedic multipliers have a higher speed, smaller area, and lower power consumption than FPGA-provided multipliers employing convolution operations. Our PEs have the added advantage of reduced memory usage.

Vedic Mathematics demonstrates low latency for multiplication and employs a unique technique based on 16 Sutras [21]. The specific method is found in Urdhva Tiryakbhyam, one of the 16 Sutras, and performs multiplication in a crosswise and vertical manner. Thus, the method allows numerical computations at a high pace by generating partial products and sums in a single iteration [22].

The Vedic multiplier architecture was used in this study, employing smaller modules to permit the multiplication of large numbers (N × N bits). We used a 2 × 2 Vedic multiplier as a fundamental building module from which higher multipliers are assembled, as shown in Figure 2. Adders were used to perform intermediate computations, and consequently optimized adder arrangements can improve multiplier efficiency [23]. Improving intermediate adder efficiency will enhance multiplier efficiency while reducing latency and size [24]. The ripple carry adder (RSA) design is a standard adder design that consists of a series of full adders. All bits must be queued, as the carry out bit of each full adder is used for the next bit operation. While this design is easy to implement, it is inefficient and slow. A carry look-ahead adder (CLAA) can compensate for the limitations of RSA by generating and propagating carries. The speed of a CLAA is balanced by the increased complexity; the number of gates and fan-in increases as the most significant digit increases. A prefix adder performs parallel operations over several stages; the Kogge–Stone adder (KSA) is an example of this type of adder. The KSA is a parallel prefix form of the CLAA and is widely considered to be the fastest adder in the industry [23]. Thus, we designed the Vedic multiplier with the KSA as shown in Figure 3.

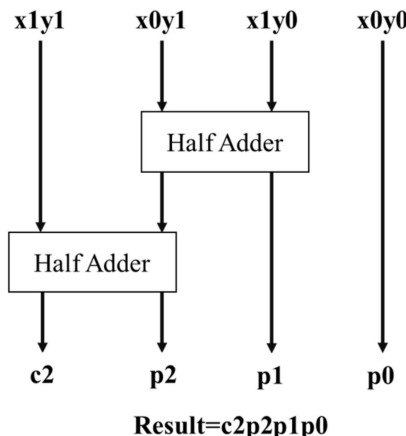

**Figure 2.** Block Diagram of 2 × 2 Vedic Multiplier.

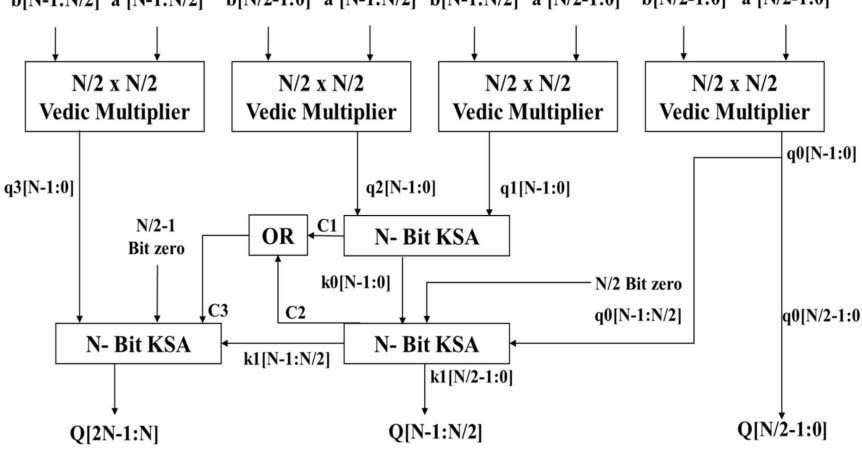

**Figure 3.** Block Diagram of Vedic multiplier using KSA.

A simulation was performed using Synopsys' Verdi simulator. The RTL simulation results for a 32 bits × 32 bits Vedic multiplier are shown in Figure 4, in which Input 1 = 36, Input 2 = 48, and Output = 1728. The synthesis was performed using a Synopsys Design Compiler. The above design was implemented in Verilog code using VCS. The RTL code was synthesized using Design Compiler in 65 nm technology. Area, speed, and power reports were compared with other methods, as listed in Table 1. Additionally, Table 2 lists the FPGA resources consumed in the implementation of the 32 × 32 multiplier. We used Vivado 2019.4 as the compiler and performed a demonstration on a ZCU102 board. Compared with the multiplier resources provided by Xilinx, there was a 31% reduction in Slice, a 16% reduction in input output block (IOB), and a 13% increase in lookup table (LUT).

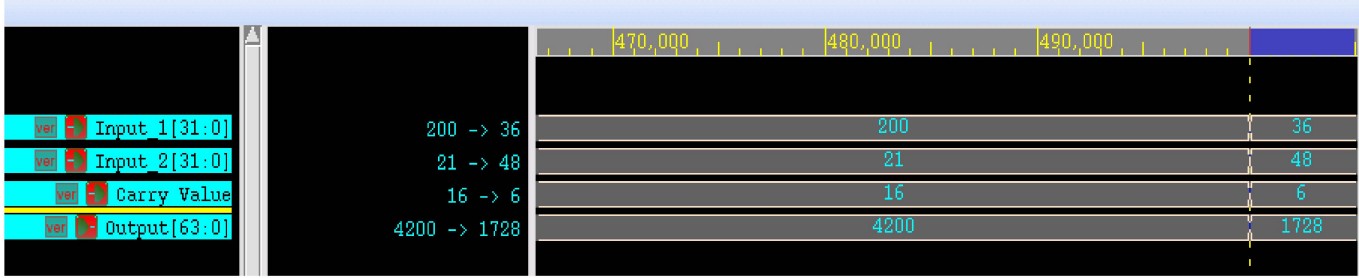

**Figure 4.** Simulation results of 32 × 32 Vedic multiplier.

**Table 1.** Analysis report of conventional multiplier and proposed Vedic multiplier.

| Parameter | Conventional Multiplier | Conventional Vedic Multiplier | Proposed Vedic Multiplier |
| --- | --- | --- | --- |
| Power (µW) | 26.351 | 27.235 | 23.33 |
| Speed (ns) | 6.213 | 5.327 | 4.1 |
| Area (µm²) | 3567 | 4283 | 3958 |

**Table 2.** Resource utilization of 32 × 32 multiplier.

| Parameter | Slice | LUT | DSP | IOB |
| --- | --- | --- | --- | --- |
| Xilinx IP | 415 | 93 | 2 | 102 |
| Ours | 287 | 107 | 0 | 86 |

### 3.3. Performance-Based Auto-Generator

Computation and communication are the two primary constraints used to improve the throughput of an accelerator. The roofline performance model quantitatively analyzes the computed throughput and required memory bandwidth for any potential solution to a convolutional neural network (CNN) design on an FPGA platform. The roofline calculation density refers to the amount of calculation required by a program per unit of memory access, with units of *FLOPs/byte*. The calculation formula is as follows:

$$I = \left( \frac{FLOPs}{byte} \right) \qquad (1)$$

The roofline model is used to evaluate the upper bound of a program's performance on hardware, and is displayed using the following Figure 5:

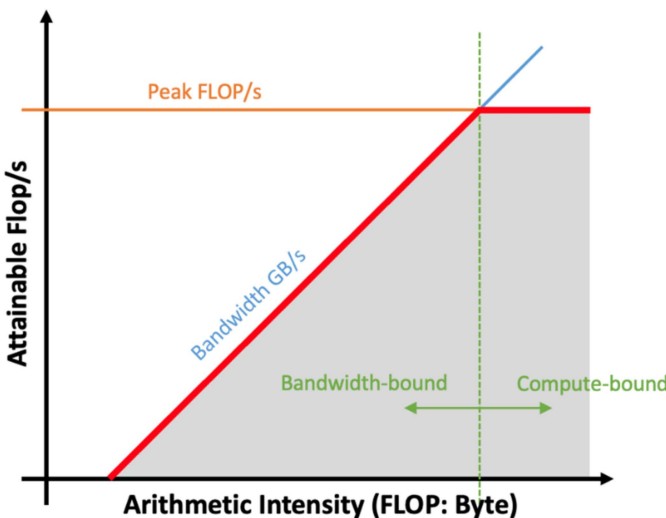

**Figure 5.** Roofline model.

When the calculation density *I* of the inference program is small, the program fetches more memory per calculation. In other words, the memory bandwidth limits the hardware design. The upper bound of program performance is expressed as a diagonal line in Figure 5, where the slope represents memory bandwidth. The greater the calculation density, the higher the upper bound of the program's achievable speed, while always using the maximum memory bandwidth. On the contrary, if the calculation density *I* is large, the program performance is limited by the maximum calculation peak of the hardware (called a calculation-intensive program). Beyond the maximum calculation peak, performance is limited by the hardware computing power, which is expressed as a horizontal line in Figure 5. The calculation speed is no longer affected by the calculation density in this regime. The calculation speed and the memory bandwidth are maximized at the intersection of the two lines; this intersection point denotes the full use of hardware resources.

Not all operators have the same computational properties in a deep learning network. That is, they may use more or less memory while performing similar calculations and are not in the same position on the roofline. Operators in deep learning networks can be classified according to computational density; conv, FC, and deconv operators are calculation-intensive operators, while ReLU, EltWise Add, Concat, etc., are memory-access-intensive operators. The same operator will also change the calculation density or change its properties owing to different parameters. For example, increasing the group or reducing the input channels of Conv will reduce the calculation density under the premise that other parameters remain unchanged.

For memory-intensive operators, the inference time has a linear relationship with memory access. By contrast, inference time has a linear relationship with calculation for computationally intensive operators. A deep learning network is often composed of a mixture of memory-intensive operators and computationally intensive operators, in which the operator attributes change. We cannot design the hardware statistically to completely leverage hardware performance.

Therefore, we propose an LSTM-based method to maximize performance-based automatic DNN structure generation. First, we implemented a method for extracting HLS features in C/C++. Then, an LSTM was designed, and the extracted features were used to predict new HLS parallel methods. We used an abstract syntax tree (AST) model to extract structure, loop nesting, variables, operators, and other specific values from HLS code. Figure 6 shows three components (parallel structure, memory access, and operands) of the vector.

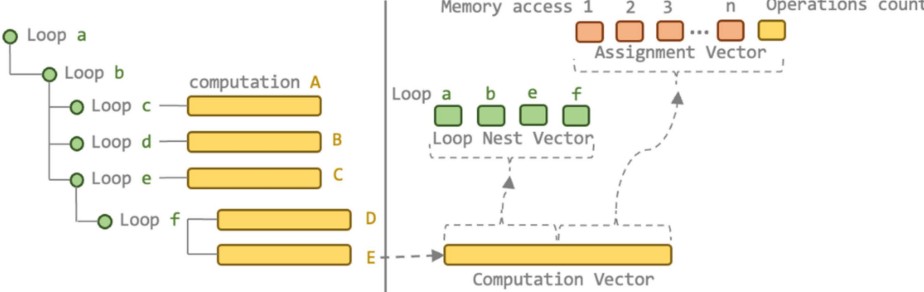

**Figure 6.** Parallel structure, memory access, and operand expression of the DNN structure.

Loop nesting codes are represented by arranging loops from the outermost to the innermost. Thus, features extracted from the loop nest can be stored as the start and end number, as shown in Figure 7.

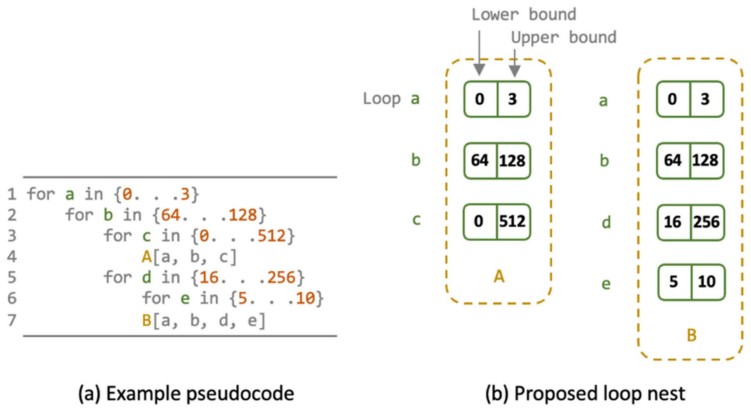

```
1 for a in {0. . .3}
2     for b in {64. . .128}
3         for c in {0. . .512}
4             A[a, b, c]
5         for d in {16. . .256}
6             for e in {5. . .10}
7                 B[a, b, d, e]
```

(a) Example pseudocode

(b) Proposed loop nest

**Figure 7.** Loop stored method.

We added tags to the loop to distinguish each loop nest level, as shown in Figure 8. Adding tags and merging cycles can avoid repeated use of loops.

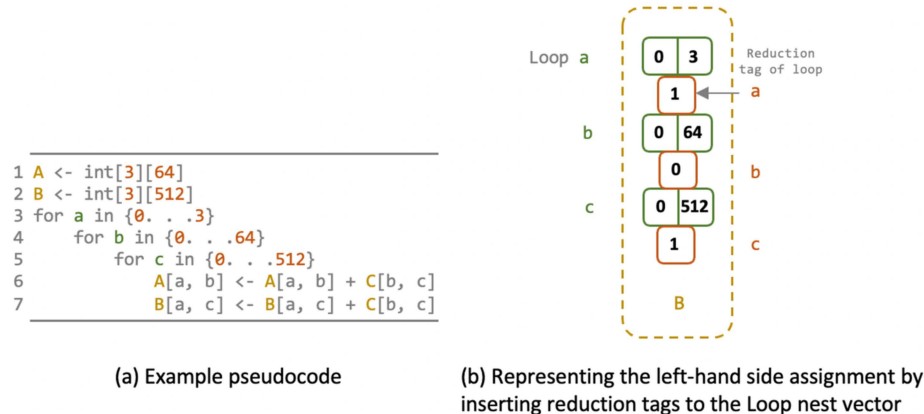

```
1 A <- int[3][64]
2 B <- int[3][512]
3 for a in {0. . .3}
4     for b in {0. . .64}
5         for c in {0. . .512}
6             A[a, b] <- A[a, b] + C[b, c]
7             B[a, c] <- B[a, c] + C[b, c]
```

(a) Example pseudocode

(b) Representing the left-hand side assignment by inserting reduction tags to the Loop nest vector

**Figure 8.** Add tag and merge loop nest.

Figures 6 and 7 show the memory access, operand expression, and loop storage of the parallel structure as parameters. In Figure 8, we characterize loop optimization by type, parameter, and point before hardware synthesis. The loop represented in the figure uses the previous loop size as the marking method because the loop level and operation affect the calculation results of the algorithm, as shown in Figure 9.

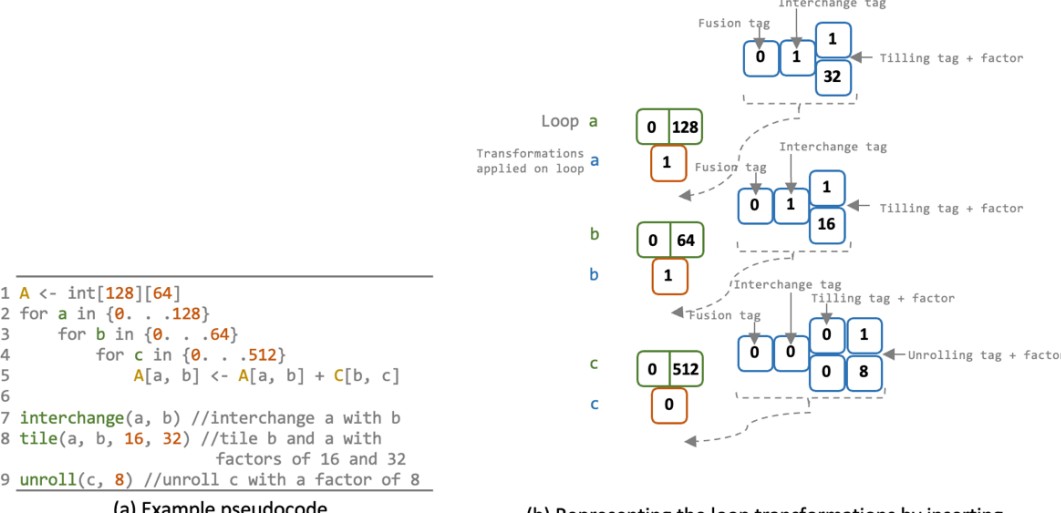

```
1 A <- int[128][64]
2 for a in {0. . .128}
3     for b in {0. . .64}
4         for c in {0. . .512}
5             A[a, b] <- A[a, b] + C[b, c]
6
7 interchange(a, b) //interchange a with b
8 tile(a, b, 16, 32) //tile b and a with
                      factors of 16 and 32
9 unroll(c, 8) //unroll c with a factor of 8
```

(a) Example pseudocode

(b) Representing the loop transformations by inserting schedule features to the Loop nest vector

**Figure 9.** Loop nest optimization pseudocode and representation.

To design an automatic scheduler, we extracted specific values (Figure 9) that determine which loop to use for Tile and Unrolling HLS pragma. As shown in Figure 10, different parallel optimizations are performed to obtain different scheduling methods in the generated HLS code. Then, the scheduler is transformed into a re-expression of the above statement through the translation method. In this process, different schedulers will synthesize various hardware designs (hardware size and latency vary between synthesized designs). These data are then used for further training via the LSTM, as shown in Figure 11.

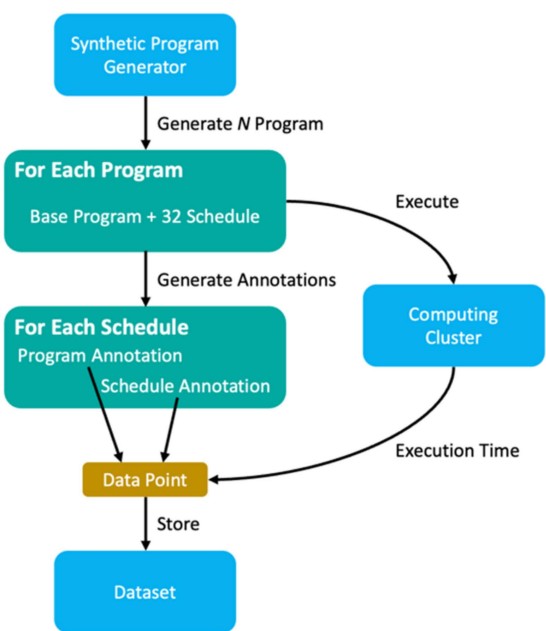

**Figure 10.** Workflow of proposed method.

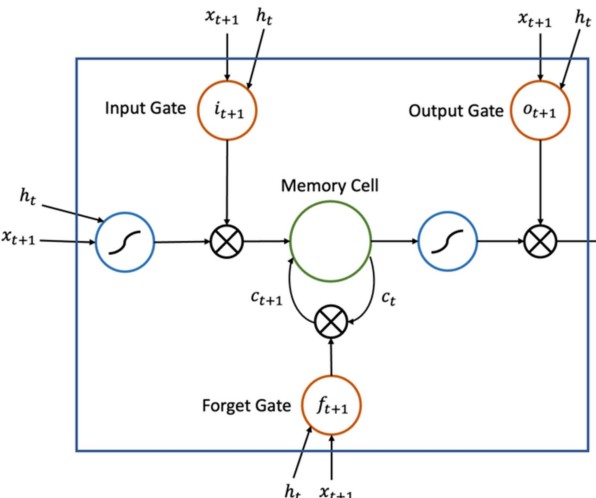

**Figure 11.** Long short-term memory training unit.

Design space exploration identifies the most optimal design with the highest throughput under the performance and resource modeling from LSTM predictions. However, the designed operating frequency is difficult to predict. Therefore, we developed a two-stage process as described in Figure 12, where we first used the proposed analysis model to filter the design space into a small set of candidates with a similar and predetermined clock frequency. Then, we generated hardware to obtain a design with the best onboard performance. The design flow framework is button-based, providing an intuitive DNN program that users write to generate an executable system on FPGA automatically. The user need not specify the nested loops of the DNN layer.

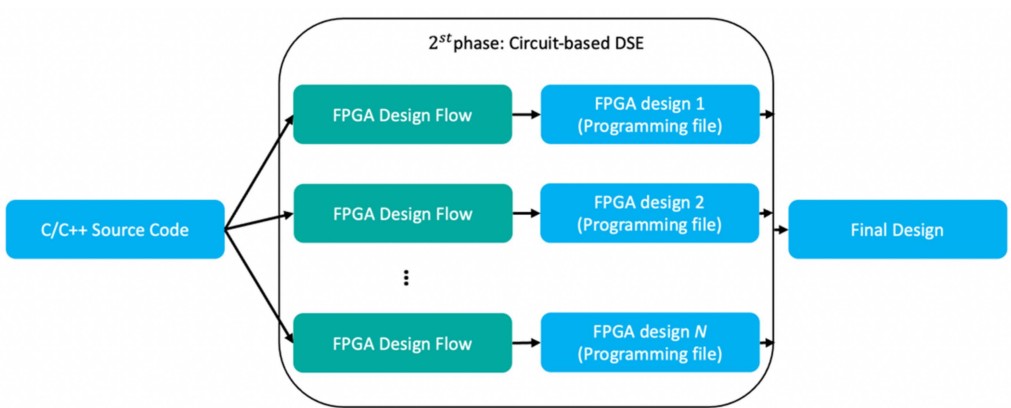

**Figure 12.** Two-stage design space exploration.

## 4. Results

We used the proposed scheme to build four DNNs: ResNet, MobileNet, DensNet, and VGG with an input size of $224 \times 224$. We compared it with the latest embedded GPUs, TX2 and TX1. We tried to use a minimum batch size owing to the real-time requirements of edge applications. We sought an example FPGA board with few gates and chose the Zynq family that was used by Oiu et al. [12] and Suda et al. [25] for comparison. Based on the experimental results summarized in Table 3, we observe that our design in Xilinx zcu104, zcu102, and Intel Cyclone V SOC (System on chip) provided higher efficiency than the TensorRT inference solution based on Nvidia Jetson TX2.

**Table 3.** Evaluation on FPGA (batch size = 1 for Fix16).

| | ZCU 104 (ms) | ZCU 102 (ms) | Cyclone V SoC (ms) | Jetson Tx1 (ms) | Jetson Tx2 (ms) |
|---|---|---|---|---|---|
| ResNet18 | 1.78 | 2.16 | 4.1 | 21 | 14.7 |
| VGG-16 | 12.9 | 15.52 | 29.1 | 151 | 105.7 |
| MobileNetV2 | 1.77 | 2.15 | 3.95 | 20.5 | 14.3 |
| DenseNet-121 | 5.67 | 6.79 | 12.8 | 66.4 | 46.4 |

We compared the accelerator design with those from two related studies in Table 4 [12,25]. We examined two aspects of performance: (1) peak CONV layer performance and (2) overall performance of all CONV layers. Our design performed significantly better than those published in previous studies.

**Table 4.** Comparison with other FPGA work.

| | Oiu [12] | Suda [25] | Ours | | |
|---|---|---|---|---|---|
| **CNN Models** | | | **VGG** | | |
| Device | Zynq 7Z45 | Stratix V GSD8 | ZCU 104 | ZCU 102 | Cyclone V SoC |
| Precision | Fixed 16 bit | Fixed 16 bit | Fixed 16 bit | Fixed 16 bit | Fixed 16 bit |
| Overall CONV GOPs | 187.8 | 136.5 | 304.2 | 253.3 | 144.7 |
| Peak CONV | 254.8 | - | 368.5 | 304.7 | 260.6 |

We used VGG16, which has 13 convolutional and three fully connected layers. Table 5 compares our proposed method with optimized CPU single instruction multiple data (SIMD) and GPU solutions. We used actual achieved performance (GOPS) as the standard measurement for fair comparison. We also used a zcu02, a zcu04, and a Kirin 970 CPU (with Arm computing library optimized) to achieve approximately 46 times the acceleration and 55 times the performance improvement.

**Table 5.** Comparison with CPU/GPU edge computing.

| Platform | CPU (SIMD) | GPU | | FPGA | | |
|---|---|---|---|---|---|---|
| Device | Kirin 970 | Jetson Tx1 | Jetson Tx2 | ZCU 104 | ZCU 102 | Cyclone V SoC |
| Precision | Float32 | Float32 | Float32 | Fixed 16 | Fixed 16 | Fixed 16 |
| Batch size | 1 | 1 | 1 | 1 | 1 | 1 |
| Latency | 720.9 | 151 | 105.7 | 12.9 | 15.52 | 29.1 |
| Speedup | 1× | 4.77× | 6.82× | 55.88× | 46.44× | 24.77× |
| Power (Watt) | 8 | 10 | 7.5 | 6.3 | 7.6 | 8.2 |

## 5. Conclusions

To meet the requirements of real-time IoT and AIoT applications, performance must be improved. Currently, cloud servers take care of the high computation requirements of complex AI applications. However, real-time computations depend on embedded equipment to avoid network delay. IoT devices are not known for powerful computations or real-time calculations. With the rising implementation of AI networks for tasks that require high precision or many variables, device localization presents a challenge. FPGA designs are the answer to these problems, as they are highly energy efficient and low cost. However, developing FPGAs is time consuming and labor intensive. In addition, they may only work within a single AI network, with further customization requiring additional time. Developers are searching for a tool that can "automatically generate" AI networks on FPGAs, thereby reducing time consumption and generating the AI networks at a hardware level. Therefore, we propose a high-performance deep learning inference optimizer with low-latency, high-throughput deployment inference for AI applications. Experimental results showed that the performance of the proposed method increased by

approximately 50 and 5 times compared with CPU SIMD (single instruction, multiple data) and GPU-based edge approaches, respectively.

**Author Contributions:** Date curation, S.J. and S.P.; Formal analysis, W.L.; Founding acquisition, Y.C.; Investigation, Y.C.; Resources, W.L.; Software, S.J. and W.L.; Writing—original draft, S.J.; Supervision, Y.C.; Validation, Y.C. All authors have read and agreed to the published version of the manuscript.

**Funding:** This work was supported by the Korea Evaluation Institute of Industrial Technology (KEIT) under the Industrial Embedded System Technology Development (R&D) Program 20016341. The EDA tool was supported by the IC Design Education Center (IDEC), Korea.

**Conflicts of Interest:** The authors declare no conflict of interest.

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
