# Peer review of "Automatic RTL Generation Tool of FPGAs for DNNs"

_electronics, doi:10.3390/electronics11030402_

Round 1

Reviewer 1 Report

This manuscript presents an automatic RTL generation flow for DNN implementation from software model to hardware. 
In particular, there are three major points that construct this work: 1. a more efficient multiplier, 2. an LSTM-based design space exploration method, and 3. an automatic generation flow.

Although the idea in this manuscript, including the new multiplier and adopting LSTM for design space exploration and adopting HLS as an intermediate representation for RTL generation, is novel and interesting, this manuscript contains too many shortcomings and needs drastic improvement to meet the scientific and professional requirements of a public journal.

Here are my detailed comments:
1. regarding automatic code generation, I cannot figure out in the current version which basic architectural units are used to construct the DNN model. The only microarchitecture I could see is the multiplier, but a single multiplier is not capable of constructing a fully functional DNN model.

2. following the above question, I also cannot clearly understand the architecture of PE and the target system. How are the layers being managed during the execution of the hardware? How are they mapped to the PEs? Is the accelerator dedicated or shared between the layers?

3. the related work looks like it has been abbreviated and there is no connection between the references to the quotes. I could hardly understand which sentence refer to which paper. This needs to be improved urgently.

4. regarding the design space exploration, what are the parameters that are being explored are not clearly listed. The authors mentioned them somehow, but I can not read them clearly from the manuscript.

5. it is quite strange that the multiplier is evaluated in an ASIC process, while the implementation target is on an FPGA that contains the special DSP for multiplication. The results of the ASIC process provide no benefit to the current content.

6. there are some inaccurate/unprofessional names, e.g. DNN reasoning, quantified, should really be DNN inference, quantized, etc.

Author Response

Automatic RTL Generation Tool of FPGAs for DNNs

Responses to Reviewer 1

We would like to thank you and the review team for seeing the potential in our paper and giving us constructive guidelines for revision. We have revised the paper thoroughly based on the comments and feedback we received. We provide a summary of the major improvements to the manuscript; we then respond to individual comments by each member of the review team.

Comment:

1) regarding automatic code generation, I cannot figure out in the current version which basic architectural units are used to construct the DNN model. The only microarchitecture I could see is the multiplier, but a single multiplier is not capable of constructing a fully functional DNN model.

Response:

1) Thank you for your advice on the problem. We do not consider work nodes that stop working correctly.

We optimized the general DNN design via the HLS code. Hence, we optimized two parts of the HLS code; firstly, we optimized the function loop architecture; secondly, we replaced the auto-generated RTL code’s DSP (generated from the HLS compiler) with our new PE design. In conclusion, the other parts of the DNN will be designed from general HLS code and the PE component will be replaced with our proposed design. (As line 58 to 63).

The whole architecture design flow as below:

We propose a space exploration model to find the optimal accelerator design that balances the DNN network's constraints and the FPGA performance. As shown in Figure 1, we generated the DNN accelerator via a 3-step procedure that included a net-work analyzer, required-performance based PE optimization, and HDL (Hardware de-scription language) auto-generation. The steps are described as follows:

  1. Developers first use a deep learning ONNX (Open Neural Network Exchange) framework to design and train the DNN target network during the design process. After training, the DNN network definition file is passed to the DNN network analyzer.
  2. Network analysis decomposes network layers from the network model, such as the conv, pooling, and FC layers. Then, it maps them to the HLS template. The net-work analyzer leaves operation logic and bit size intact in the conv, pooling, and FC layers to ensure that logic remains as designed.
  3. In the third step, the LSTM-based generator automatically designs the optimized DNN HLS.

Figure 1. A diagram of the proposed system.

Comment:

2) following the above question, I also cannot clearly understand the architecture of PE and the target system. How are the layers being managed during the execution of the hardware? How are they mapped to the Pes? Is the accelerator dedicated or shared between the layers?

Response:

2) As response 1, We replaced the auto generated RTL code’s DSP, which is generated by a general HLS compiler, with our PE design.

Comment:

3) the related work looks like it has been abbreviated and there is no connection between the references to the quotes. I could hardly understand which sentence refer to which paper. This needs to be improved urgently.

Response:

3) Thanks for your suggestion. We deleted the four references below:

  1. Chui, K.T.; Gupta, B.B.; Vasant, P. A Genetic Algorithm Optimized RNN-LSTM Model for Remaining Useful Life Prediction of Turbofan Engine.Electronics 2021, 10, 285. https://doi.org/10.3390/electronics10030285.
  2. Pirahandeh, M.; Ullah, S.; Kim, D.H. 2021. A Distributed Edge-Based Scheduling Technique with Low-Latency and High-Bandwidth for Existing Driver Profiling Algorithms. Electronics, 10(8), p.972. https://doi.org/10.3390/electronics10080972.
  3. Alom, M.Z.; Taha, T.M.; Yakopcic, C.; Westberg, S.; Sidike, P.; Nasrin, M.S.; Hasan, M.; Van Essen, B.C.; Awwal, A.A.S.; Asari, V.K. A State-of-the-Art Survey on Deep Learning Theory and Architectures. Electronics 2019, 8, 292. https://doi.org/10.3390/electronics8030292.
  4. Zheng, Y.; Yang, H.; Jia, Y.; Huang, Z. PermLSTM: A High Energy-Efficiency LSTM Accelerator Architecture. Electronics 2021, 10, 882. https://doi.org/10.3390/electronics10080882.

In addition, we focus reference on chapter 2 which mainly discuss DNN FPGA design as below (As line 65 to 89).

  1. Aydonat et al. showed that the Winograd-based solution could be implemented on FPGA, reducing the need for multiplication in a DNN [9]. New techniques such as quantization [12,13] and model compression [14,15] are also used in the FPGA DNN accelerator to reduce the size of the DNN model and diminish latency in the DNN inference process. Automation tools to quickly deploy DNNs into FPGAs have been researched by other studies [16]. The DLL used a unified computing unit (for example, a fixed size computing engine), which yields reduced computing performance when compared to a dedicated design for different networks [17,18].

Comment:

4) regarding the design space exploration, what are the parameters that are being explored are not clearly listed. The authors mentioned them somehow, but I can not read them clearly from the manuscript.

Response:

4) We are exploring memory access, operand expression and loop storage of parallel structure parameters with DNN HLS code. (As line 201 to 219).

As explanation in line 220-221, We add ‘In Figure 6 and Figure 7, we demonstrate memory access, operand expression and loop storage of parallel structure as parameters.’.

Figure 6. Shows the parallel structure, memory access, and operand expression of the DNN structure.

Figure 7. Shows the Loop Stored Method.

Comment:

5) it is quite strange that the multiplier is evaluated in an ASIC process, while the implementation target is on an FPGA that contains the special DSP for multiplication. The results of the ASIC process provide no benefit to the current content.

Response:

5) We replace the auto-generated RTL code DSP, which was generated by a general HLS compiler, with our PE design. DSP exhibits power consumption issues, and with quantized DNN operation, DSP which is design by 32bit cannot be fully used. Hence, we compare our N-bit PE with Xilinx IP. As shown, the proposed multiplier has superior power, speed, and board area.

Comment:

6) there are some inaccurate/unprofessional names, e.g. DNN reasoning, quantified, should really be DNN inference, quantized, etc.

Response:

6) Thank you for your advice regarding the problem. As you pointed out, we used unprofessional names. All instances have been changed to 'DNN inference' and 'quantized' as below. (Please see line 73, 74, 78 and 110.)

Studies involving platform-based DNN search methods [10,11] improve upon previously published results. However, the authors only consider DNN inference delay on the CPU and GPU, neglecting DNN inference delay on FPGAs.

New techniques such as quantization [12,13] and model compression [14,15] are al-so used in the FPGA DNN accelerator to reduce the size of the DNN model and diminish latency in the DNN inference process. However, these methods may be limited by the DNN model itself, as not all DNN models can be compressed. The compressed DNN model also does not necessarily meet the real-time performance constraints of target AIOT platforms.

We optimized the DNN design in two ways. First, we redesigned the PE so that energy consumption is optimized in the automatically generated neural network. The proposed PE also supports N-bit calculations, permitting automatic quantization of the network into N-bits. Second, we implemented feature value extraction in the HLS C/C++ inference code and used this feature value to train LSTM. This automatically generates the DNN-RTL design with the highest degree of parallelism.

Also, we attach a certificate of English editing from Editage.

Reviewer 2 Report

Specific comments
- line 14: please spell out RTL or open the acronym in the abstract
- line 19: please spell out LSTM or open the acronym in the abstract
- please spell out all the acronyms in the paper the first time you use them, not below in in the manuscript
- In the Tables PAPAMETER is PARAMETER instead?
Please also explain why you did use the Zynq family of the Xilinx instead of the Virtex UltraScale+ that feature larger LUT, CLBs, DSPs numbers in general
- lines 42-54: you basically invite the users to exploit HLS to convert DNNs to RTL code. Personally I suggest to not write this as a recommendation but only as an option. Surely it is not easy to design HDLs - VHDL or Verilog - at the lever of RTL but, in case the designers have the skills, they might gain better performance in terms of FPGA resource utilization. In other words, if you can cope with RTL design you might fit larger DNNs in the same FPGA device w.r.t. the DNN the HLS can fit. Can you comment please?

Author Response

Automatic RTL Generation Tool of FPGAs for DNNs

Responses to Reviewer 2

We would like to thank you and the review team for seeing the potential in our paper and giving us constructive guidelines for revision. We have revised the paper thoroughly based on the comments and feedback we received. We provide a summary of the major improvements to the manuscript; we then respond to individual comments by each member of the review team.

Comment:

1) line 14: please spell out RTL or open the acronym in the abstract

Response:

1) We added Register-Transfer Level, the full name of RTL, to the abstract as below. (Please see line 14.)

RTL (Register-transfer Level) programming, hardware verification, and precise resource allocation are needed to build a high-performance FPGA accelerator for DNNs.

Comment:

2) line 19: please spell out LSTM or open the acronym in the abstract

Response:

2) We added Long Short-Term Memory, the full name of LSTM, to the abstract as below. (Please see line 19.)

We also introduce an LSTM (Long Short-Term Memory) based model to predict performance and automatically generate a DNN model suiting the developer requirements.

Comment:

3) please spell out all the acronyms in the paper the first time you use them, not below in in the manuscript

Response:

3) Thank you for your advice on the problem. We opened first time used acronyms and added them all. (DNN: line 11, GPU: line 23, CPU: line 30, HLS: line 58, HDL: line 95, ONNX: line 99, IOB: line 160, LUT: line 160, SIMD: line 286.) And if the previously opened acronyms were repeated, it was deleted. (LSTM: line 50, HLS: line 103.)

Comment:

4) In the Tables PAPAMETER is PARAMETER instead?

Please also explain why you did use the Zynq family of the Xilinx instead of the Virtex UltraScale+ that feature larger LUT, CLBs, DSPs numbers in general

Response:

4) In Tables 1 and 2, PARAMETER is correct. Thanks for pointing that out.

The reasons for using the Zynq family are as follows. Reason 1: Zynq was used in the reference papers [12][26], making the family a good comparison. Reason 2: Our goal is to target boards with fewer gates. We added an explanation for this decision as below. (Please see line 253-254.)

We sought an example FPGA board with few gates and chose the Zynq family that was used in Oiu et al. [12] and Suda et al. [26] for comparison.

Comment:

5) lines 42-54: you basically invite the users to exploit HLS to convert DNNs to RTL code. Personally I suggest to not write this as a recommendation but only as an option. Surely it is not easy to design HDLs - VHDL or Verilog - at the lever of RTL but, in case the designers have the skills, they might gain better performance in terms of FPGA resource utilization. In other words, if you can cope with RTL design you might fit larger DNNs in the same FPGA device w.r.t. the DNN the HLS can fit. Can you comment please?

Response:

5) Thank you for your advice on the problem. We added ‘As DNN structure changes rapidly, it is difficult for the hardware design of DNN to keep up with the software design of DNN. Therefore, our proposition allows users to perform resource allocation and optimization during RTL design when deploying DNN on FPGAs.’

As described above, we think DNN design can see efficiency improvements with proper RTL design. However, as the structure of the DNN changes rapidly, it is difficult for the hardware design to keep up with software design. For example, in the case of the yolo series, representative structures are rapidly published; yolov4, yolov5, and yolox. Hence, it is expected that HLS design will improve DNN performance. (Please see line 43-46.)

Round 2

Reviewer 1 Report

Although I have given a rejection in the first round, this manuscript is given a major revision by other reviewers, hence, I hope the following issues can be addressed if there is a following round of revision.

  1. A diagram for the generated system architecture should be added, otherwise, the system design space exploration will not be able to provide any optimal output. The authors can refer to the following paper which adopts HLS and also generates the DNN accelerators. https://dl.acm.org/doi/10.1145/3289602.3293915 
  2. The figures and tables should be well aligned. 
  3. The fonts of the words in the figures should be improved, it is not clear with a greyscale printout.

Author Response

Automatic RTL Generation Tool of FPGAs for DNNs

Responses to Reviewer 1

We would like to thank you and the review team for seeing the potential in our paper and giving us constructive guidelines for revision. We have revised the paper thoroughly based on the comments and feedback we received. We provide a summary of the major improvements to the manuscript; we then respond to individual comments by each member of the review team.

Comment:

1) A diagram for the generated system architecture should be added, otherwise, the system design space exploration will not be able to provide any optimal output. The authors can refer to the following paper which adopts HLS and also generates the DNN accelerators. 

Response:

1) Thank you for your advice on the problem. We do not consider work nodes that stop working correctly. We modified and added Figure 1 as shown below.

Figure 1. Proposed system.

Comment:

2) The figures and tables should be well aligned. 

Response:

2) Thanks for your suggestion. We have realigned the figures and tables correctly. The aligned figure is Figure 4, and the table is Table 3, 4, and 5. We aligned as below.

Figure 4. Simulation results of 32 x 32 Vedic Multiplier.

Table 3. Evaluation on FPGA (batch size = 1 for Fix16).

ZCU 104

(ms)

ZCU 102

(ms)

Cyclone V SoC

(ms)

Jetson Tx1

(ms)

Jetson Tx2

(ms)

ResNet18

1.78

2.16

4.1

21

14.7

VGG-16

12.9

15.52

29.1

151

105.7

MobileNetV2

1.77

2.15

3.95

20.5

14.3

DenseNet-121

5.67

6.79

12.8

66.4

46.4

Table 4. Comparison with other FPGA work.

Oiu[12]

Suda[26]

Ours

CNN models

VGG

Device

Zynq 7Z45

Stratix V GSD8

ZCU 104

ZCU 102

Cyclone V SoC

Precision

Fixed 16 bit

Fixed 16 bit

Fixed 16 bit

Fixed 16 bit

Fixed 16 bit

Overall CONV GOPs

187.8

136.5

304.2

253.3

144.7

Peak CONV

254.8

-

368.5

304.7

260.6

Table 5. Comparison with CPU/GPU edge computing.

Platform

CPU (SIMD)

GPU

FPGA

Device

Kirin 970

Jetson Tx1

Jetson Tx2

ZCU 104

ZCU 102

Cyclone V SoC

Precision

Float32

Float32

Float32

Fixed 16

Fixed 16

Fixed 16

Batch size

1

1

1

1

1

1

Latency

720.9

151

105.7

12.9

15.52

29.1

Speedup

1x

4.77x

6.82x

55.88x

46.44x

24.77x

Power(Watt)

8

10

7.5

6.3

7.6

8.2

Comment:

3) The fonts of the words in the figures should be improved, it is not clear with a greyscale printout.

Response:

3) Thank you for your advice regarding the problem. As you pointed out, we have modified the fonts in the figures so that you can clearly see them and improve the picture quality. The figure we modified is Figure 7, 8, 9, and 12, and the modified figure is as below.

Figure 7. Loop stored method.

Figure 8. Add Tag and Merge Loop Nest.

Figure 9. Loop nest optimization pseudocode and representation.

Figure 12. Two stage design 

Reviewer 2 Report

I see you addressed the previous comments. In fact, I still suggest reconsidering the use of Virtex families instead of Zynq for future studies. The explanation is however acceptable.

Author Response

Automatic RTL Generation Tool of FPGAs for DNNs

Responses to Reviewer 2

We would like to thank you and the review team for seeing the potential in our paper and giving us constructive guidelines for revision. We have revised the paper thoroughly based on the comments and feedback we received. We provide a summary of the major improvements to the manuscript; we then respond to individual comments by each member of the review team.

Comment:

1) I see you addressed the previous comments. In fact, I still suggest reconsidering the use of Virtex families instead of Zynq for future studies. The explanation is however acceptable.

Response:

1) Thank you for your suggestion and understanding. For the suggestion, we will proceed with future work.

Round 3

Reviewer 1 Report

All my comments are addressed.